# A New Feature with the Potential to Detect the Severity of Obstructive Sleep Apnoea via Snoring Sound Analysis

**DOI:** 10.3390/ijerph17082951

**Published:** 2020-04-24

**Authors:** Shota Hayashi, Meiyo Tamaoka, Tomoya Tateishi, Yuki Murota, Ibuki Handa, Yasunari Miyazaki

**Affiliations:** 1Department of Respiratory Medicine, Graduate School of Medical and Dental Sciences, Tokyo Medical and Dental University (TMDU), Tokyo 113-8510, Japan; hayagend@tmd.ac.jp (S.H.); tateishi.pulm@tmd.ac.jp (T.T.); miyazaki.pilm@tmd.ac.jp (Y.M.); 2Department of Respiratory Physiology and Sleep Medicine, Graduate School of Medical and Dental Sciences, Tokyo Medical and Dental University (TMDU), Tokyo 113-8510, Japan; 3Yamaha Corporation Electronic Devices Division, Shizuoka 438-0192, Japan; yuki.murota@music.yamaha.com (Y.M.); ibuki.handa@music.yamaha.com (I.H.)

**Keywords:** obstructive sleep apnoea, snoring, sound analysis

## Abstract

The severity of obstructive sleep apnoea (OSA) is diagnosed with polysomnography (PSG), during which patients are monitored by over 20 physiological sensors overnight. These sensors often bother patients and may affect patients’ sleep and OSA. This study aimed to investigate a method for analyzing patient snore sounds to detect the severity of OSA. Using a microphone placed at the patient’s bedside, the snoring and breathing sounds of 22 participants were recorded while they simultaneously underwent PSG. We examined some features from the snoring and breathing sounds and examined the correlation between these features and the snore-specific apnoea-hypopnea index (ssAHI), defined as the number of apnoea and hypopnea events during the hour before a snore episode. Statistical analyses revealed that the ssAHI was positively correlated with the Mel frequency cepstral coefficients (MFCC) and volume information (VI). Based on clustering results, mild snore sound episodes and snore sound episodes from mild OSA patients were mainly classified into cluster 1. The results of clustering severe snore sound episodes and snore sound episodes from severe OSA patients were mainly classified into cluster 2. The features of snoring sounds that we identified have the potential to detect the severity of OSA.

## 1. Introduction

Obstructive sleep apnoea (OSA) is a prevalent sleep disorder that occurs when a patient stops continuously breathing because of upper airway collapse. It causes oxygen desaturation and arousal, which interrupts normal sleep. A complete blockage of the upper airway is called apnoea, whereas a partial blockage is called hypopnea [1]. The average number of apnoea and hypopnea events per hour of sleep is used to define an apnoea-hypopnea index (AHI). AHI values for adults are categorized as 5 ≤ AHI < 15, mild; 15 ≤ AHI < 30, moderate; and ≤ 30, severe. OSA patients have an increased risk for cardiovascular diseases, hypertension, diabetes, stroke, neurocognitive deficits, daytime sleepiness, depression, and mood disorders [2,3].

The gold standard of OSA diagnosis is overnight polysomnography (PSG) [4]. PSG monitors many physical functions, including brain activity (electroencephalography (EEG)), eye movement (electro-oculography (EOG)), muscle activity or skeletal muscle activation (electromyography (EMG)), and cardiac rhythms (electrocardiography (ECG)). To monitor these body functions, the patient is monitored by more than 20 sensors. In addition, the financial cost to the patient is high, and the patient must stay at the sleep clinic or hospital for one night because of the requirement for complex PSG equipment. Therefore, the patient cannot sleep as they usually do. These complicated devices impose stress on not only patients but also clinical staff because they need to spend a long time connecting the patient to the equipment. For these reasons, although there are many patients with potential sleep apnoea, treatments for patients with potential sleep apnoea have not progressed [5].

Typical noninvasive treatments for OSA are continuous positive airway pressure (CPAP) and the use of an oral appliance (OA). CPAP, which supplies a continuous flow of pressurized air through a nose mask, has been internationally recognized as an effective treatment for OSA [6]. OA, which holds the patient’s lower jaw forward, has been widely recognized as an effective treatment for mild to moderate OSA and suggested as an alternative treatment for patients who cannot tolerate or refuse CPAP [7,8,9].

To properly treat OSA patients with these devices, efficient screening tools are needed. Early recognition and prompt, appropriate intervention reduces mortality and improves the quality of life of OSA patients. Although PSG is the current gold standard for the diagnosis of OSA, the cumbersome process and a prolonged waiting period for this examination may affect the patient’s sleep and delay timely treatment [5].

To overcome these drawbacks, some methods for detecting OSA other than PSG have been studied [10], and several snoring sound analytic programs have been developed because snoring is the most common symptom of OSA [11]. Snoring sounds are caused by the vibration of soft tissue in the narrowed upper airways, including anatomical structures such as the soft palate, uvula, and pharynx [12,13]. Some of these methods have been developed using features obtained from snore sounds, such as Mel frequency cepstral coefficients (MFCC), formant frequencies (FF), non-Gaussianity scores, loudness, and pitch [3]. These methods for detecting OSA are becoming available, but most remain at the prototype stage and are insufficiently validated.

As the demand for OSA diagnosis increases, simpler, contactless, and inexpensive devices for sleep testing are desired. To develop such a system, we aimed to illustrate the ability of new features computed from snore sounds to classify the severity of OSA. The novelty of our approach is that we use a simple microphone to collect snore sounds from a whole-night audio recording without attaching sensors to the patient. Moreover, we computed a new feature from each snore episode and proposed a new objective score for quantifying snore intensity.

## 2. Materials and Methods

### 2.1. Acquisition of PSG Data and Snore Sounds

Twenty-two subjects (11 males and 11 females with a mean age of 64.4 ± 12.0 years, mean body mass index of 26.7 ± 5.7, mean AHI of 38.4 ± 23.4) with suspected OSA were referred for a PSG test at the Tokyo Medical and Dental University Medical Hospital, Tokyo, Japan. Routine PSG tracings were made using clinical devices (SOMNOscreen^TM^ Plus, Fukuda Denshi, Tokyo, Japan). Subjects reported to the hospital at 7 p.m. and were discharged at 7 a.m. the following morning. PSG scoring was performed by a trained technician. Apnoeas and hypopneas were scored according to the American Academy of Sleep Medicine criteria [14].

Snoring and breathing sounds were recorded using a microphone (AT898, audio-technica, Tokyo, Japan) that was hanging approximately 30 cm above the patient’s head and a recorder (DR-44WL, TEAC CORPORATION, Tokyo, Japan) with a 16-bit sampling rate of 44,100 Hz.

### 2.2. Snore Database

Our clinical laboratory technicians extracted each snore episode from the data recorded overnight and confirmed the time when each snore episode started. Thus, we enumerated whole-snore sound waveforms, and a total of 10,078 snore episodes were obtained from 22 subjects (Figure 1).

### 2.3. Snore-Specific AHI

We defined the snore-specific AHI (ssAHI) as the number of apnoea and hypopnea events during the hour before a snore episode. We hypothesized that calculating the ssAHI in the immediate vicinity of the snore episode more accurately reflects the relationship between snoring sounds and the severity of OSA (Figure 2). In defining ssAHI, we used AHI as a reference and figured that one hour would reflect the events within the same sleep cycle. For each snoring sound, ssAHI was calculated, and a time interval of one hour was chosen to indicate how many apnoea/hypopnea occurrences each snoring sound was equivalent to in the most recent hour.

### 2.4. Feature Extraction

From each snore episode, we determined the different features: MFCC, FF, and volume information (VI) (Table 1).

MFCC is the dominant feature used for speech recognition. Our speech sounds and snore sounds have similarities because both are generated by regions of the upper airway anatomy. The steps involved in the extraction of the MFCC features are as follows: Sound signal is first processed through a pre-emphasis filter in order to emphasize the higher frequency components. Next, short-time frames of sound signal are created using overlapping Hamming windows. Typically, the duration of the analysis window is 20–30 ms with an overlap of 50%. This is followed by deriving the frequency domain representation for each of the short-time frames. Discrete Fourier transform is used for this purpose. The phase information is discarded from the resulting short-term spectrum. The magnitude or the power spectrum is then warped to the Mel-scale using a set of nonlinearly spaced filters. The Mel-filter bank is a set of triangular Mel-weighted filters. Next, logarithmic compression is performed followed by the application of discrete cosine transform to derive a set of de-correlated cepstral coefficients. Finally, a low-time liftering operation is performed to discard the higher-order coefficients. In the context of automatic speech recognition, only the first 13 coefficients are retained and they are collectively known as MFCC features.

A formant is a concentration of acoustic energy around a particular frequency in the speech wave. The formant with the lowest frequency is called F1, the second F2, and the third F3. F1–F3 correspond to the stricture of the pharynx, tongue advancement, and lip rounding [15]. We regarded the FF from snoring and breathing sounds as the acoustic characteristics of the upper airway during apnoea, hypopnea, and snoring. Hence, we computed F1, F2, and F3 from the snore episodes.

VI refers to the volume of snoring sounds and changes in snoring volume. We defined a snore episode waveform as a short wave (SW) (Figure 3), and the waveform including the last 10 s before the end of a SW was defined as a long wave (LW) (Figure 4). From the SW, LW, and root mean square (RMS), we computed VI1–VI7. VI1 is the maximum RMS value in every frame. VI2 is the RMS value in every frame of each SW. VI3 is the average frame RMS value of the SW. VI4 is the dispersion of the frame RMS values of the SW. VI5 is the RMS value in every frame of each LW. VI6 is the average frame RMS value of the LW. VI7 was the dispersion of the frame RMS values of the LW (Table 2).

### 2.5. Statistical Analysis and Clustering

We used the Pearson correlation coefficient *r* to measure the correlation of the MFCC, FF, and VI with the ssAHI. A *p* value < 0.05 was considered statistically significant.

After the statistical analysis, k-means clustering was performed using the correlated features.

## 3. Results

Among the subjects, there were 11 males and 11 females, with a mean age of 64.4 (standard deviation (SD) 12.0) years. The mean body mass index was 26.7 (SD 5.7) kg/m^2^, and the mean AHI was 38.4 (SD 23.4) events/hour (Table 3).

A total of 10,078 snore episodes were obtained from the data recorded overnight from the 22 subjects, and three types of features—MFCC, FF, and VI—were computed.

The results of the correlation analysis of these features and the ssAHI are presented in Table 4, Table 5 and Table 6.

The MFCC was positively correlated with the ssAHI (*r* = 0.33, *p* < 0.05) (Table 4), and VI also showed a positive correlation with the ssAHI (*r* = 0.28, *p* < 0.05) (Table 5). The FF and ssAHI showed only a weak correlation, as the value for the correlation coefficient was less than 0.2 (*p* < 0.05) (Table 6).

The results of clustering mild snore sound episodes (5 ≤ ssAHI < 15) and severe snore sound episodes (30 ≤ ssAHI) with the MFCC and VI are shown in Table 7. A total of 1334 mild snore sound episodes were classified as belonging to cluster 1, and 3193 severe snore sound episodes were classified as belonging to cluster 2. The results of clustering the snore sound episodes obtained from the mild OSA patients (5 ≤ AHI < 15) and the snore sound episodes obtained from the severe OSA patients (30 ≤ AHI) with the MFCC and VI are also shown in Table 7. A total of 1091 snore sound episodes obtained from the mild OSA patients were classified as belonging to cluster 1, and 3777 snore sound episodes obtained from the severe OSA patients were classified as belonging to cluster 2.

## 4. Discussion

In this study, we investigated a method for analyzing patient snore sounds to detect the severity of OSA as an alternative to PSG. Unlike recent studies [16,17], we did not use any special equipment to record snore sounds. Some of the other studies [4,18,19,20] used a tracheal microphone or a microphone embedded in a facemask. However, these microphones are often uncomfortable for subjects. In our case, we used a noncontact microphone hanging approximately 30 cm above the patient’s head. This method can record snore sounds without any physical connections to the subjects and does not disturb subjects’ sleep.

We aimed to detect the severity of OSA with snoring sound analysis and used the MFCC, FF, and VI. Although the MFCC and FF were also used in past studies, they were not adequate for determining the severity of OSA [3,5,21]. U. R. Abeyratne et al. used some of these features to investigate snore sounds for OSA/non-OSA classifications [22]. Shahin Akhter et al. also used these features to detect the presence of OSA during non-rapid eye movement (NREM) and rapid eye movement (REM) sleep [3]. These authors detected OSA/non-OSA classifications and the presence of OSA during NREM and REM sleep at some level. However, they were not able to determine the severity of OSA. To solve this problem, we defined the ssAHI as a new objective score for quantifying snore intensity and computed the VI as a new feature of snore sounds. The AHI is represented by the average number of apnoea and hypopnea events per hour of sleep. That is, the AHI is calculated by dividing the total number of overnight apnoea and hypopnea events by the sleep time. Therefore, the AHI does not represent the severity of apnoea and hypopnea when each snoring sound is occurring because the AHI is the average of apnoea and hypopnea events per hour overnight. For this reason, it was necessary to use a value correlating each snore sound with apnoea severity, and we defined the ssAHI as the number of apnoea and hypopnea events during the hour before a snore episode.

We assumed that the ssAHI in the immediate vicinity of a snore episode more accurately reflected the relationship between each snoring sound and the severity of OSA, regardless of the AHI, which represents the average of the apnoea and hypopnea events per hour of the whole night.

To investigate the characteristics of each snore sound episode in detail, we also computed the VI as a new feature from each snore episode to more clearly capture the sequential change in the snoring sound. In the past studies, these features were mainly used for analyzing the timbre and volume of the snoring sound and did not focus on the fine temporal changes in the snoring sound, and some studies analyzed only selected snores (usually loud events) [3,17]. However, we analyzed all snore sound episodes and computed their VI. In past studies, it was difficult to distinguish whether silence was due to upper airway collapse or sleeping without apnoea or hypopnea. Shahin Akhtter et al. stated that one of the limitations was that their technique could not be applied to patients who did not snore because people without OSA did not have a snoring noise or the volume was low. To overcome this limitation, we analyzed all snore sound episodes and computed the VI. VI1, VI2 and VI3 represent the volume of snoring sounds and have large values in more severe OSA patients. VI4 represents the dispersion of snoring sounds and has a large value in more severe OSA patients. VI5 and VI6 are the volumes, including during the time before the snore sound begins, and they have smaller values in more severe OSA patients. VI7 represents the dispersion of the volume, including the time before the snore sound begins, and has a large value in OSA patients. With these features, it is now possible to perform snore sound analysis even if there is a silent period immediately before snoring.

In this study, we found that the ssAHI had a weak correlation with the MFCC and VI (Table 4 and Table 5). Using the features MFCC and VI, we found that snore sound episodes can be broadly classified into mild and severe types (Table 7). Mild snore sound episodes (5 ≤ ssAHI < 15) and snore sound episodes in mild OSA patients (5 ≤ AHI < 15) were mainly classified into cluster 1. Severe snore sound episodes (15 ≤ ssAHI) and snore sound episodes in severe OSA patients (15 ≤ AHI) were mainly classified into cluster 2. Although it was not clear what clinical indicators (e.g., sex and BMI) each of clusters 1 and 2 were closely related to, it was significant that we were able to broadly divide mild OSA patients and severe OSA patients, and mild snore sound episodes and severe snore sound episodes into two clusters. Increasing the accuracy of this clustering will allow us to use snore sounds to determine the severity of the patient’s OSA.

The index we used, named the ssAHI, could be determined by calculating the number of times apnoea and hypopnea occurred one hour before the snore sound occurred. This index may help to determine the severity of OSA from sequential changes in snore sounds in the future.

There are some limitations of PSG, which is the gold standard for OSA diagnosis, such as its high cost, complex equipment, and inconvenience. Additionally, there are no studies showing that the severity of OSA can be diagnosed by the magnitude and frequency of snore sounds. Based on our proposed method, simpler, contactless, and inexpensive OSA screening methods using snore sounds can be developed.

The limitation of our study is the small number of samples and sensitivity to individual differences. Further, it was not possible to analyze what clinical indicators were associated with each of the clusters 1 and 2. This was influenced by the individual differences in the snore sounds among the subjects. We consider that by increasing the number of subjects, reducing the effect of the individual differences in snoring sounds, and improving features, we can more rigorously divide them into two clusters. 

## 5. Conclusions

Our study demonstrates that the ssAHI has a weak correlation with the MFCC and VI. Moreover, OSA patients can be broadly classified into mild and severe types based on the MFCC and VI. Our results suggest that the features of the snoring sounds we defined may reflect the severity of OSA. Using these features, we found that snore sound episodes can be broadly classified into mild and severe types. As the correlation between the feature values and the AHI is weak, it is necessary to improve the features so that the correlation is enhanced and not affected by the individual differences in snoring noises.

## Figures and Tables

**Figure 1 ijerph-17-02951-f001:**
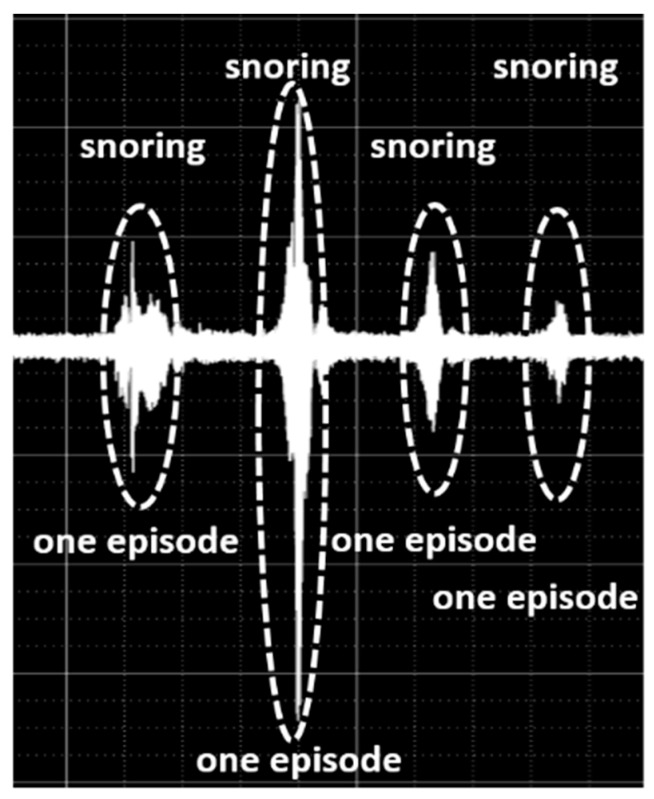
Enumerating sound waveforms of 10,078 snore episodes.

**Figure 2 ijerph-17-02951-f002:**
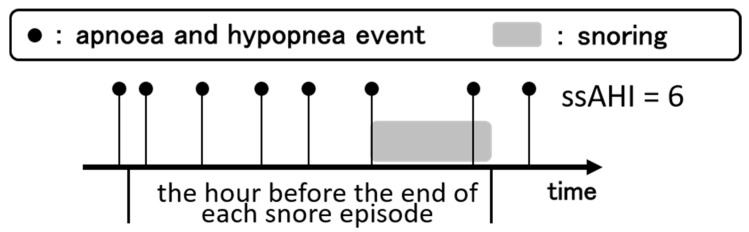
The snore-specific apnoea-hypopnea index (ssAHI) was defined as the number of apnoea and hypopnea events during the hour before a snore episode.

**Figure 3 ijerph-17-02951-f003:**
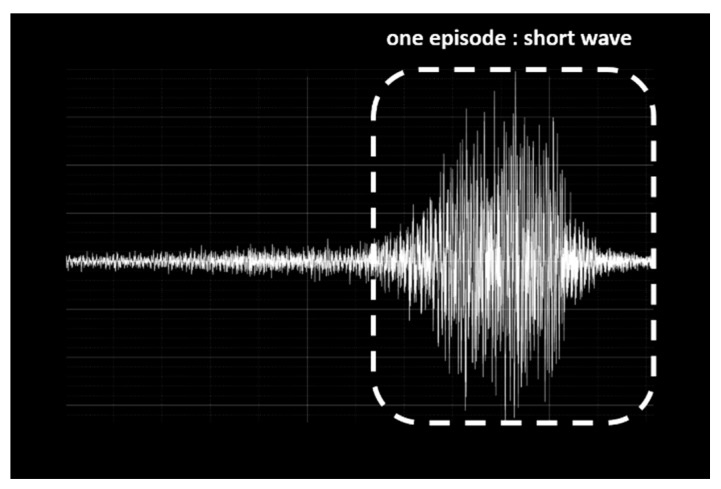
A short wave (SW) is defined as one episode.

**Figure 4 ijerph-17-02951-f004:**
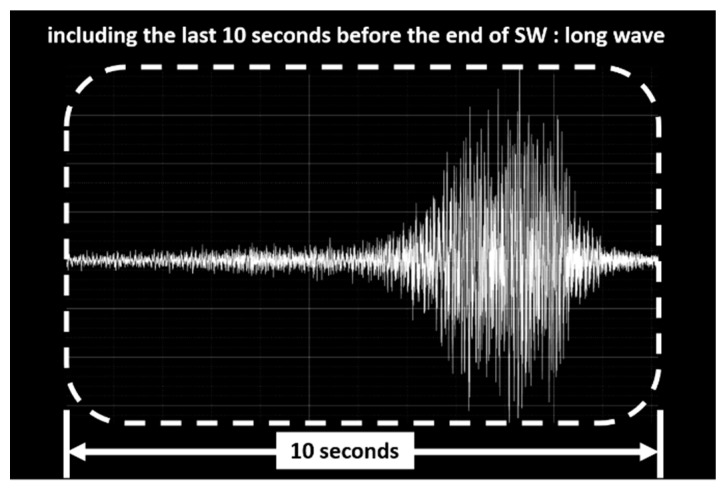
A long wave (LW) is defined as including the last 10 s before the end of the short wave (SW).

**Table 1 ijerph-17-02951-t001:**
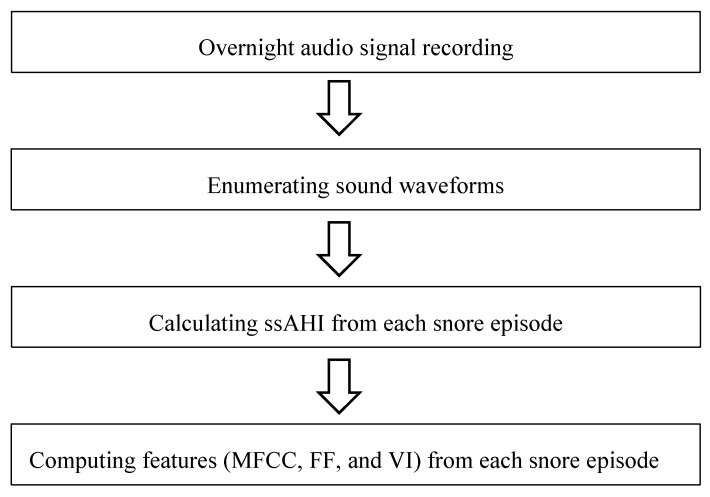
Flow chart of feature extraction.

**Table 2 ijerph-17-02951-t002:** Details of volume information (VI). (SW, short wave; LW, long wave; RMS, root mean square; VI, volume information.).

SW	one snore episode waveform
LW	one waveform including the last 10 seconds
RMS	root mean square
VI1	the maximum RMS value in the every frame RMS value of each SW
VI2	the RMS value in every frame of each SW
VI3	the average of frame RMS values of the SW
VI4	the dispersion of frame RMS values of the SW
VI5	the RMS value in every frame of each the LW
VI6	the average of frame RMS values of the LW
VI7	the dispersion of frame RMS values of the LW

**Table 3 ijerph-17-02951-t003:** Subject characteristics. Data are presented as the mean ± SD or n. (BMI, body mass index; AHI, apnoea-hypopnea index).

Sex (M/F)	11/11
Age (yr)	64.4 ± 12.0
BMI (kg/m^2^)	26.7 ± 5.7
AHI (events/hour)	38.4 ± 23.4

**Table 4 ijerph-17-02951-t004:** The results of correlation analysis for Mel frequency cepstral coefficients (MFCC) and the ssAHI. (MFCC, Mel frequency cepstral coefficients; ssAHI, snore-specific AHI.)

mfcc_1	0.3309
mfcc_2	−0.09966
mfcc_3	−0.14657
mfcc_4	0.186239
mfcc_5	−0.1558
mfcc_6	0.008772
mfcc_7	0.014088
mfcc_8	0.011975
mfcc_9	−0.14602
mfcc_10	−0.01543
mfcc_11	−0.18824
mfcc_12	−0.09601
mfcc_13	−0.15889

**Table 5 ijerph-17-02951-t005:** The results of the correlation analysis for the VI and the ssAHI. (VI, volume information; ssAHI, snore-specific AHI).

volinfo_1	0.249201
volinfo_2	0.273147
volinfo_3	0.283421
volinfo_4	0.057741
volinfo_5	0.238167
volinfo_6	0.00525
volinfo_7	0.254416

**Table 6 ijerph-17-02951-t006:** The results of the correlation analysis for formant frequency (FF) and the ssAHI. (FF, formant frequencies; ssAHI, snore-specific AHI.)

1st formant	0.154177
2nd formant	0.079543
3rd formant	0.087834

**Table 7 ijerph-17-02951-t007:** The results of clustering mild and severe snore sound episodes with the MFCC and VI. The results of clustering mild obstructive sleep apnoea (OSA) patients and severe OSA patients with the VI. (MFCC, Mel frequency cepstral coefficients; VI, volume information.).

	Cluster 1	Cluster 2
mild snore sound episodes (5 ≤ ssAHI < 15)	1334	586
severe snore sound episodes (30 ≤ ssAHI)	1911	3193
snore sound episodes from mild OSA patients (5 ≤ AHI < 15)	1091	191
snore sound episodes from severe OSA patients (30 ≤ AHI)	2313	3777

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
