# Peer review of "A New Feature with the Potential to Detect the Severity of Obstructive Sleep Apnoea via Snoring Sound Analysis"

_ijerph, 2020, doi:10.3390/ijerph17082951_

Round 1

Reviewer 1 Report

The manuscript: “A new feature with the potential to detect the severity of obstructive sleep apnoea via snoring sound analysis” evaluates a really important topic. The authors found ssAHI as the novel index, and revealed that the ssAHI was correlation with MFCC and VI. I have some concerns about this manuscript, as below.

Page 3:
Please describe how to analysis MFCC more detail. Especially, it is difficult to understand what mfcc_1 to mfcc_13 are showing in Table 4.

Page 7:
Please describe more detail in cluster 1 and 2 in table 7. It may be better to note what each cluster represent. For example, figure 1 and table 2 legend in below paper is easy to understand.

Bailly S, et al. Obstructive Sleep Apnea: A Cluster Analysis at Time of Diagnosis. PLoS One. 2016 Jun 17;11(6):e0157318.

REFERENCES:
Reference 7 & 8 were a little old. Please refer new one or systematic review, as below;

Aarab G, Lobbezoo F, Hamburger HL, Naeije M. Oral appliance therapy versus nasal continuous positive airway pressure in obstructive sleep apnea: a randomized, placebo-controlled trial. Respiration 2011;81:411–9.

Ramar K, Dort LC, Katz SG, Lettieri CJ, Harrod CG, Thomas SM, Chervin RD. Clinical Practice Guideline for the Treatment of Obstructive Sleep Apnea and Snoring with Oral Appliance Therapy: An Update for 2015. J Clin Sleep Med. 2015 Jul 15;11(7):773-827.

Reviewer 2 Report

  1. Why was 1 hour period selected as suitable to define a new indicator ssAHI. Does this have any physiological justification?
  2. Does the proposed method have a chance of being automated if the recorded acoustic signal had to be pre-developed by a professional technician? What about recognition of acoustic events that are not related to the subject of the study?
  3. Do the low correlation coefficients presented above entitle to draw conclusions about the usefulness of the method in comparison with other studies? Shouldn't more experiments be conducted to confirm the proposed method?

The answers to these questions are important to me from the point of view of the factual assessment of the presented article and should be included in its content.

Round 2

Reviewer 2 Report

I accept your answers. I still find the idea of a new indicator interesting, but not supported by sufficient clinical trials. I leave the final decision on whether or not to allow the work to be printed in the hands of the magazine's editors.